# Bacterial Resistance against Heavy Metals in *Pseudomonas aeruginosa* RW9 Involving Hexavalent Chromium Removal

**Fatini Mat Arisah** [1], **Amirah Farhana Amir** [1], **Norhayati Ramli** [1,2], **Hidayah Ariffin** [1,2], **Toshinari Maeda** [3], **Mohd Ali Hassan** [1] **and Mohd Zulkhairi Mohd Yusoff** [1,2,*]

1   Department of Bioprocess Technology, Faculty of Biotechnology and Biomolecular Sciences, Universiti Putra Malaysia, Serdang 43400, Selangor, Malaysia; fatiniarisah@gmail.com (F.M.A.); amirah98farhana@gmail.com (A.F.A.); yatiramli@upm.edu.my (N.R.); hidayah@upm.edu.my (H.A.); alihas@upm.edu.my (M.A.H.)
2   Institute of Tropical Forestry and Forest Products (INTROP), Universiti Putra Malaysia, Serdang 43400, Selangor, Malaysia
3   Department of Biological Functions Engineering, Graduate School of Life Science and Systems Engineering, Kyushu Institute of Technology, 2-4 Hibikino, Wakamatsu-ku, Kitakyushu 808-0196, Japan; toshi.maeda@life.kyutech.ac.jp
*   Correspondence: mzulkhairi@upm.edu.my; Tel.: +60-03-97698060

**Abstract:** *Pseudomonas aeruginosa* RW9 is a promising candidate for the bioremediation of chromium hexavalent (Cr(VI)) pollution, as it resists a high concentration of up to 60 mg/L of Cr(VI). Leaving cells exposed to Cr(VI) has large bioreduction potential, implying its capacity to extract the ions from the contaminated medium. In this study, the tolerance for and distribution of Cr(VI) were investigated to identify the cells' adaptation and removal strategies. Micro-characterization analysis was conducted to assess the effect of Cr(VI) on the cells. The cells' elongation was observed at higher Cr(VI) concentrations, signifying their adaptation to DNA damage caused by Cr(VI) toxicity. Cr(VI) distribution analysis showed that the strain developed a complex mechanism to adapt to Cr(VI), based on surface-bound (0.46 mg/L), intracellularly accumulated (1.24 mg/L) and extracellular sequestration (6.74 mg/L), which accounted for 85% of the removal efficiency. The extracellular sequestration might be attributable to the production of metabolites, in accordance with the fourier-transform infrared spectroscopy (FTIR) spectra and orcinol analysis that confirmed the presence of a glycolipid biosurfactant, rhamnolipid. Remarkably, the rhamnolipid was slightly induced in the presence of Cr(VI). From the data obtained, it was confirmed that this local strain is well equipped to survive high doses of Cr(VI) and has great potential for application in Cr(VI) bioremediation.

**Keywords:** *Pseudomonas* sp.; biosurfactant; heavy metals; chromium; bioremediation

## 1. Introduction

Heavy metal pollution has been a great concern around the globe. The compounds often exist as elements that possess the non-degradability characteristic and persist in the environment over time [1]. Various remediation tools have been developed to remediate the pollutants. Conventional methods such as physical adsorption [2], coagulation [3] and membrane filtration [4] are among the commonly used techniques to treat metals from industrial effluent. However, such technologies have contributed to sludge generation as secondary pollution [5]. Apart from that, additional chemicals such as precipitation and coagulation agents raise environmental and safety concerns [6]. As for membrane filtration, the presence of other compounds or particulate matter can cause the membrane to be clogged and need to be replaced constantly [7]. To curb the problem, pre-treatments need to be applied to the wastewater prior to subjection to a membrane filtration system, which causes the technique to become financially inefficient [5]. These drawbacks have caused researchers to shift to the development of microbial remediation as a potential alternative to conventional techniques [7].

Microbial remediation using living bacterial strains has been recognized as a promising and efficient method for treating pollution [7]. In this technique, living strains, often isolated from contaminated sites, are utilized to remediate the target pollutants. Several strains have been reported to remove ionic metals successfully [8], with those among the most efficient being from the genus of *Pseudomonas* [9], which is ubiquitous in the environment. The strains exhibit several heavy-metal-resistant genes, which explain their tolerance to the extreme environment [10] via various adaptation mechanisms [11]. These mechanisms include intracellular sequestration, localization within specific organelles, metallothionein binding, particulate metal accumulation, extracellular precipitation, complex formation, and extracellular sequestration by membrane-bound metabolites such as reductase and biosurfactant [12].

In addition to such adaptations, which involve proteomic changes, some of the strains respond to the unfavourable environment by excreting biosurfactants. A biosurfactant is a surface-active material produced by microorganisms that is classified according to its molecular weight [13,14]. In *P. aeruginosa*, the biosurfactant produced is of low molecular weight and an anionic glycolipid polymer. However, the role of the metabolite was described as strain-dependent. Some previous studies reported a significant role of biosurfactants in promoting metal remediation, while others claimed otherwise. The compound was found to assist in metal remediation by promoting extracellular sequestration while preventing the ionic metals from breaching into the cells [13]. Meanwhile, some other work reported its sole role as a protective barrier to the cells and did not report it to contribute to the former function [14].

The key to understanding the adaptation strategies of microbial strains lies in the distribution of the pollutants, which provides an overview of the metabolic paths enabling microbes to survive and persist in heavy-metal-contaminated environments. Thus, in this study, *P. aeruginosa* RW9 tolerance was first assessed by subjecting the cells to a range of Cr(VI) concentrations. The chromium hexavalent (Cr(VI)) was chosen as the model, considering the seriousness of its pollution of the environment. This is attributed to its application in various industries such as leather tanning, steel production, wood preservation and textiles [7]. In addition, the ion has also been found to be carcinogenic and teratogenic. The application of bioremediation is one of the green technologies employed in environmental pollution.

Previous work on *P. aeruginosa* showed its remarkable ability to degrade long-chain fatty acids, which can be attributed to its glycolipid biosurfactant. The surface-active metabolite was reported as an excellent emulsifier for a wide range of hydrophobic pollutants. However, the role of biosurfactants in metal remediation is ambiguous, as some studies have reported contradictory findings. Previous researchers have reported a role in metal complexation, while others have suggested otherwise, claiming that its sole function was as a protective barrier to the cells. The barrier helps in shielding the cells from the direct effect of the ions, this suggests that the role of the biosurfactant was strain-dependent and highly related to its characteristics. Thus, in this study, we investigated the potential use of *P. aeruginosa* RW9 against Cr(VI) as the heavy metal model for its mechanism of uptake and effects of the metal ions on biosurfactant synthesis. The findings of this study may deliver theoretical provision for the bioremediation of metal-contaminated environments and support a potential practical application.

## 2. Materials and Methods

### 2.1. Bacterial Strain and Growth Culture Conditions

The local biosurfactant-producing strain used in this study was previously isolated from the compost of a palm oil empty fruit bunch [15]. The identification and characterization were performed using Biolog Gen III, with physical and chemical characterization. The strain was identified as *Pseudomonas aeruginosa* RW9 [15]. It was kept at $-80\ ^\circ$C in a glycerol medium as a stock culture. The strain was routinely streaked onto a nutrient agar plate and incubated at 30 $^\circ$C. For inoculum preparation in the Cr(VI) bioremediation

experiment, a single colony of the 24 h incubated strain was cultured into 50 mL of nutrient broth (NB) (peptone, 5 g/L; yeast extract, 3 g/L) and placed in a shaking incubator at 150 rpm.

### 2.2. Determination of Cr(VI) Tolerance by Cell Viability Measurement

All the glassware used was washed with 60% ($v/v$) nitric acid and rinsed with deionized water to remove any possibility of interference by other metals before the experimental procedures [16]. A stock solution of 1 g/L Cr(VI) was prepared by dissolving $K_2Cr_2O_7$ in deionized water before being autoclaved at 121 °C for 20 min. The stock solution was then added into NB at concentrations ranging from 10 to 160 mg/L, with a twofold dilution interval [17]. To observe the strain tolerance against Cr(VI), the growth was quantified based on the optical density using a spectrophotometer (Hitachi UV 2900). For the growth assessment, 10% ($v/v$) inoculum with an optical density (OD600) between 1.8 and 2.0 was inoculated in 45 mL of NB with Cr(VI). The mixture was incubated at 30 °C and mixing at 150 rpm for 24 h. The culture without Cr(VI) served as control. To eliminate the effect of metabolite produced, samples taken out at predetermined time intervals were centrifuged (10,000× $g$) for 10 min at 4 °C before being re-suspended in NB for optical density measurement. The specific growth rate (μ) was determined from the growth profile obtained by using the following equation below [18]:

$$\mu = \frac{\Delta \ln OD}{\Delta t} \tag{1}$$

The effect of Cr(VI) on the morphological changes of the cells was assessed by using a field emission scanning electron microscope (FESEM). At the end of the incubation period, the cells were harvested by centrifugation at 10,000× $g$ for 10 min at 4 °C and prepared as suggested by [19] with some modifications. The cell pellet obtained was fixed using 2.5% glutaraldehyde for 3–4 h at 4 °C. After fixation, cells were washed with 0.1 M sodium cacodylate buffer and post-fixed with 2% osmium tetroxide for 2 h. Then, the cells were washed with the same buffer another three times and subsequently dehydrated in 10, 30, 50, 70 and 90% absolute ethanol sequentially for 10 min in each drying process instead of acetone. This is because ethanol has a higher boiling point (79 °C) relative to acetone (57 °C), which promotes slower moisture loss and prevents drastic disordering in lipid-containing cell structures [20]. This prevents the structure of the cells from appearing as dented, which can affect the image produced. The samples were placed into a critical point drying basket before being subjected to critical point drying for 90 min. The dried, dehydrated samples were mounted onto aluminium stubs and further gold-coated before being viewed using a FESEM (JEOL JSM 7600F, Tokyo, Japan).

### 2.3. Chromium (Cr(VI)) Detection and Removal Efficiency

The Cr(VI) concentration was determined using the HACH kit (ChromaVer Powder Pillow, HACH, Loveland, CO, USA), which uses the 1,5-diphenylcarbazide method [21]. The reagent was added to 1 mL of each sample (cell-free supernatant). The samples were swirled and left for 5 to 10 min for colour development. Purple-violet colouration is formed, indicating the presence of Cr(VI). The intensity of the colouration was measured at an absorbance of 540 nm by using a UV spectrophotometer. The percentage of chromium removal (%) was calculated based on Equation (2) [22]:

$$\text{Removal (\%)} = \frac{(\text{initial } [Cr(VI)] - [Cr(VI)] \text{ removed at predetermined time interval})}{\text{initial } [Cr(VI)]} \times 100 \tag{2}$$

To determine the mechanisms involved in Cr(VI) removal, samples were pipetted out at each time interval from 4 h until 24 h of incubation. The bacterial suspensions were centrifuged at 10,000× $g$ and 4 °C for 20 min, to obtain the supernatant, which was labelled as residual. Then, the pellets were washed with 1 mL of 10 mM EDTA solution twice and re-suspended to desorb the ions from the cell surface. The suspensions were re-centrifuged

(10,000× *g*) at 4 °C for 10 min. The supernatant collected from this treatment was labelled as surface-bound. Later, the remaining cell pellets were washed twice and re-suspended with 1 mL of 1 N $HNO_3$. Subsequently, the cells were lysed by sonication (Vibra Cell VCX 750, Newtown, USA) at 50 MHz on ice for 20 min. The cell suspensions were then centrifuged at 10,000× *g* and 4 °C for 10 min, and the obtained supernatant was labelled as intracellular. To obtain the concentration of extracellular sequestered Cr(VI), both the surface-bound and intracellular Cr(VI) concentrations were subtracted from the residual. The formulae below (Equations (3) and (4)) were used for the calculation. The detailed steps involved to obtain the samples for each treatment phase are illustrated in Figure S1.

$$Cr(VI)_{residual} = Cr(VI)_{extracellular} + Cr(VI)_{surface\text{-}bound} + Cr(VI)_{intracellular} \tag{3}$$

$$Cr(VI)_{extracellular} = Cr(VI)_{residual} - Cr(VI)_{surface\text{-}bound} - Cr(VI)_{intracellular} \tag{4}$$

A HACH kit (Chromium Total Powder Pillow, HACH, Loveland, CO, USA) was used to confirm the presence of Cr(III) by following Method 8024. This procedure used the alkaline hypobromite oxidation method [21], in which the trivalent chromium (Cr(III)) in the sample was oxidized to the hexavalent form by the hypobromite ion under alkaline conditions. The oxidation process was conducted by mixing and swirling the samples in the Chromium 1 Reagent Powder Pillow, Chromium 2 Reagent Powder Pillow and Acid Reagent Powder Pillow, sequentially, with an incubation period of 5 min for each reagent. The total chromium content was determined by the 1,5-diphenylcarbazide method by mixing ChromaVer 3 Chromium Reagent Powder Pillow with the samples and incubating for another 5 min. The intensity of the violet colouration was measured at an absorbance of 540 nm by using a UV spectrophotometer. The Cr(III) was determined by subtracting the results of a separate hexavalent chromium test from the results of the total chromium test. Medium without cells served as the negative control to explicate its role in Cr(VI) remediation.

### 2.4. Quantification of Biosurfactant

Samples labelled as residual (Section 2.3) were tested using the orcinol reagent to detect the presence of a biosurfactant [23]. The intensity of the yellow colouration formed as the reagent was added to the samples was measured at an absorbance of 421 nm (Hitachi UV 2900, Tokyo, Japan) [23]. Commercial rhamnolipid (Sigma-Aldrich, Saint Louis, MO, USA) was used to construct a standard curve for biosurfactant quantification. All the results obtained are represented as the mean ± standard deviation.

### 2.5. Biosurfactant Extraction and Characterization

For characterization analysis, fermentation was carried out in a 2 L flask with 10% inoculum in 1 L of NB with 10 mg/L Cr(VI), and medium without Cr(VI) served as a control. At the end of the incubation time, the biosurfactant was extracted by using acid precipitation coupled with solvent extraction as suggested by [23]. Briefly, the whole broth was centrifuged at 10,000× *g* for 10 min to remove the cells before being acidified using concentrated HCl to pH 3. The supernatant was then transferred to a separating funnel, and ethyl acetate at a 1:1 ratio was added for the separation process. The mixture was shaken vigorously and allowed to form two separate layers; the upper layer was the extracted biosurfactant (in ethyl acetate), while the bottom layer was the NB. This process was repeated three times to ensure all the biosurfactant was extracted out from the NB. The top aqueous layer was transferred to a separate flask before adding an additional 0.5 g of magnesium sulphate for every 100 mL of ethyl acetate to remove traces of water. The mixture was then filtered and evaporated by using a rotary evaporator to yield a brown gum extract. The extract was then freeze-dried before subjecting it to Fourier transform infrared spectroscopy (FTIR) analysis. The functional groups present in the biosurfactant were identified using a diamond single reflection attenuated total reflectance (FTIR-ATR) device (Model N4200, Jasco, Tokyo, Japan). The spectra were measured between the wave

numbers 4000 and 400 cm$^{-1}$ at a spectral resolution of 4 cm$^{-1}$ and compared to the spectra of commercial rhamnolipid [24].

## 3. Results and Discussion

### 3.1. Cr(VI) Tolerance of Pseudomonas aeruginosa RW 9

In the present study, it was found that *P. aeruginosa* RW 9 can grow well and tolerate up to 40 mg/L Cr(VI), with an OD value ranging from 1.70 to 1.67 as compared to the control (1.71 ± 0.20). However, at a concentration of 80 mg/L, the OD (0.90 ± 0.06) was found to be greatly reduced and was considered as showing total inhibition at a concentration of 160 mg/L (OD, 0.33 ± 0.10). Considering the result obtained, the growth profiling of *P. aeruginosa* RW 9 was further examined at a concentration of Cr(VI) ranging between 0 and 60 mg/L as depicted in Figure 1. Despite the approximate OD value obtained at the end of incubation for Cr(VI) with concentrations of 0–40 mg/L, the growth profile obtained was fairly different. It was observed that cells exhibited slower growth relative to the control (μ: 0.63 h$^{-1}$) as the Cr(VI) concentration increased from 10 to 20 and 40 mg/L, with μ of 0.54 h$^{-1}$, 0.43 h$^{-1}$ and 0.39 h$^{-1}$, respectively. Towards the end of cultivation, the OD of cells at 20 and 40 mg/L was found to increase steadily after a slight drop at 12 h. At the 12th hour of incubation, at which the exponential phase was entered, the cells might have started to produce a secondary metabolite that helped them to thrive in an unfavourably higher Cr(VI) concentration, resulting in the increment of OD observed [18]. At 50 and 60 mg/L, the final OD600 were 0.62 and 0.60, respectively, which were about 70% lower than those of the control. The pattern observed might be attributed to severely damaged biomolecules and DNA as a response to chromium toxicity at high concentrations [25]. The result obtained was in agreement with the previous findings on the pattern of bacterial growth of a Pseudomonas strain reported by [26].

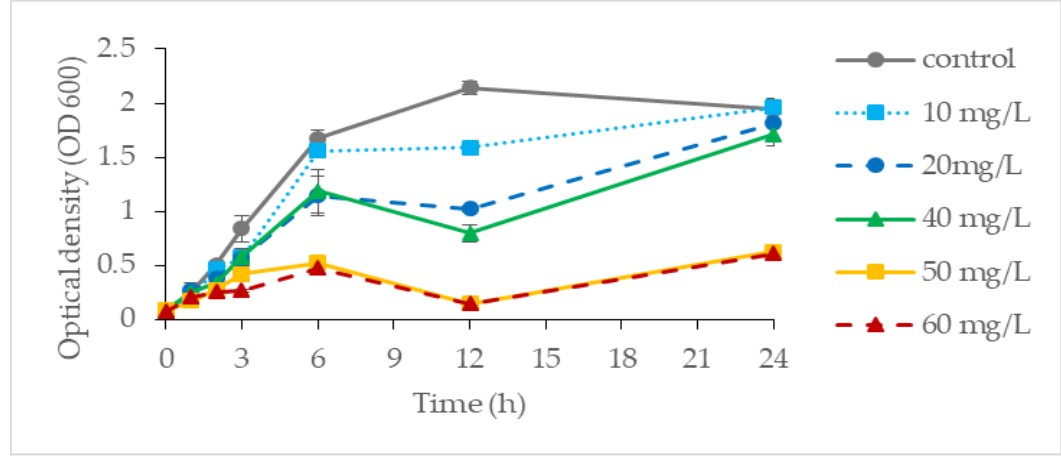

**Figure 1.** The growth profile of *P. aeruginosa* RW 9 at different Cr(VI) concentrations in NB.

To further confirm the effect of Cr(VI) on the cells, FESEM analysis was conducted to compare the morphological changes of the cells in the control experiment to those of those cultivated in higher initial concentrations of Cr(VI) (Figure 2). It was found that the cells in the control experiment were multitudinous compared to the cells in higher Cr(VI) concentrations. It was also observed that, at high Cr(VI) concentrations, the cells seemed to exhibit a rough surface, which was attributed to the adsorbed Cr ions. Similar findings were reported by [21,27]. In addition, the cells also appeared to be elongated at higher Cr(VI) concentrations. According to [27], cells elongated into filaments as the result of continual longitudinal growth without cell division. The ability of cells to divide was abolished by DNA damage due to the effect of chromium toxicity, which resulted in the induction and expression of the division inhibitor SulA.

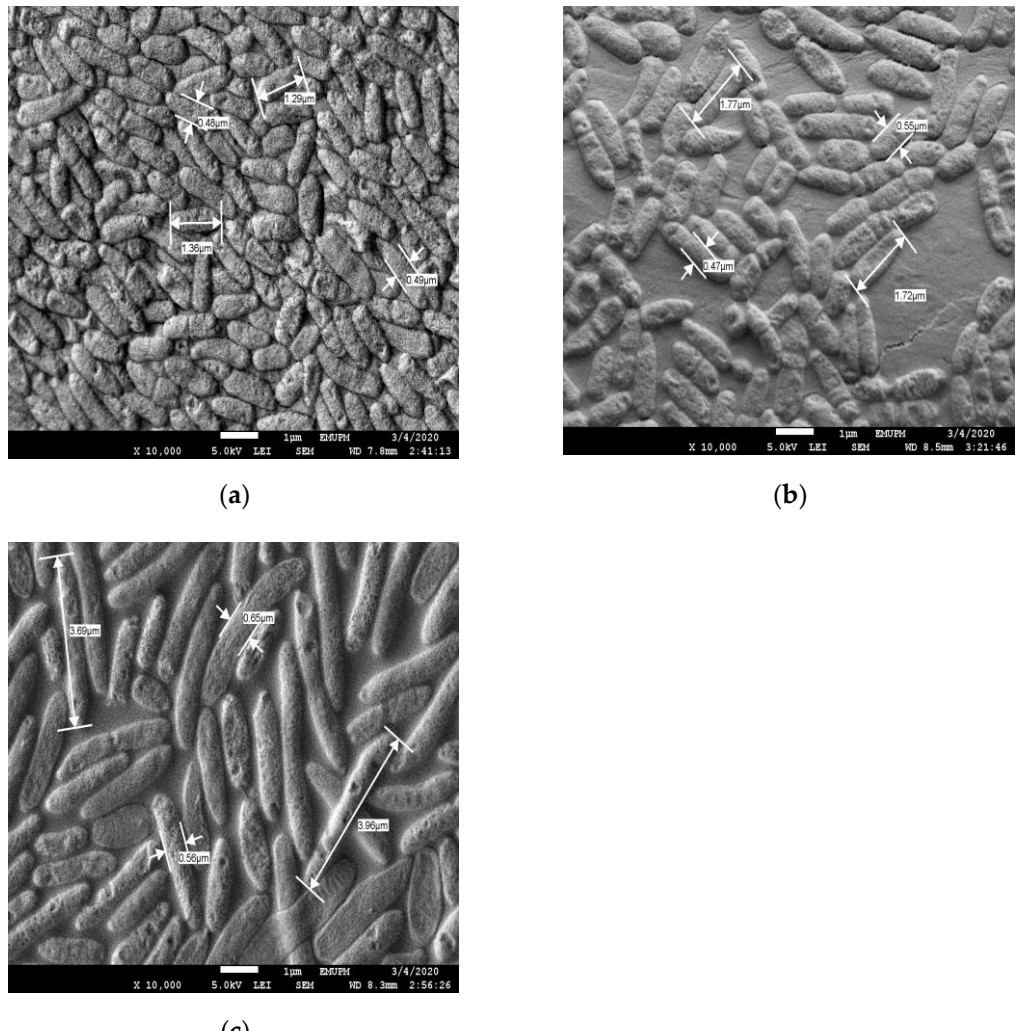

**Figure 2.** FESEM image of *P. aeruginosa* RW 9 after 24 h fermentation at 10,000× magnification: (**a**) Medium without Cr(VI) (control); (**b**) Medium with 10 mg/L Cr(VI); (**c**) Medium with 160 mg/L Cr(VI).

### 3.2. Chromium Removal Mechanisms

The highest removal efficiency was obtained for 10 mg/L, with a lesser effect on the cell growth (Figure 3). Thus, this concentration was chosen to further investigate the potential removal mechanism of *P. aeruginosa* RW 9, which was about 100 times the permissible Cr(VI) concentration for wastewater effluent [28]. It was observed that the Cr(VI) was reduced as early as 4 h, with a dominant mechanism of extracellular sequestration (Figure 4). The sequestration was speculated to be enabled by the production of membrane-associated products such as an extracellular enzyme (reductase) and biosurfactant, as depicted in Figure 5. These metabolites are predominant in most Pseudomonads [29]. The more the metabolites that were produced, the lesser the amount of Cr(VI) being adsorbed onto the cells. The secreted metabolites readily reduced and/or sequestered Cr(VI) to Cr(III) extracellularly. This prevented the penetration of the ions into the cell membrane since Cr(III) cannot readily diffuse into the cell membrane as Cr(VI) ions. On another occasion, it was reported that the presence of an underlining biosurfactant decreased the amount of pollutants adsorbed on the bacterial cell wall, thus protecting the bacteria from the toxicity [30]. The increasing trend of the extracellular Cr(VI) obtained was in accordance with the gradual decrease in the intracellular and surface-bound Cr(VI) concentration over time.

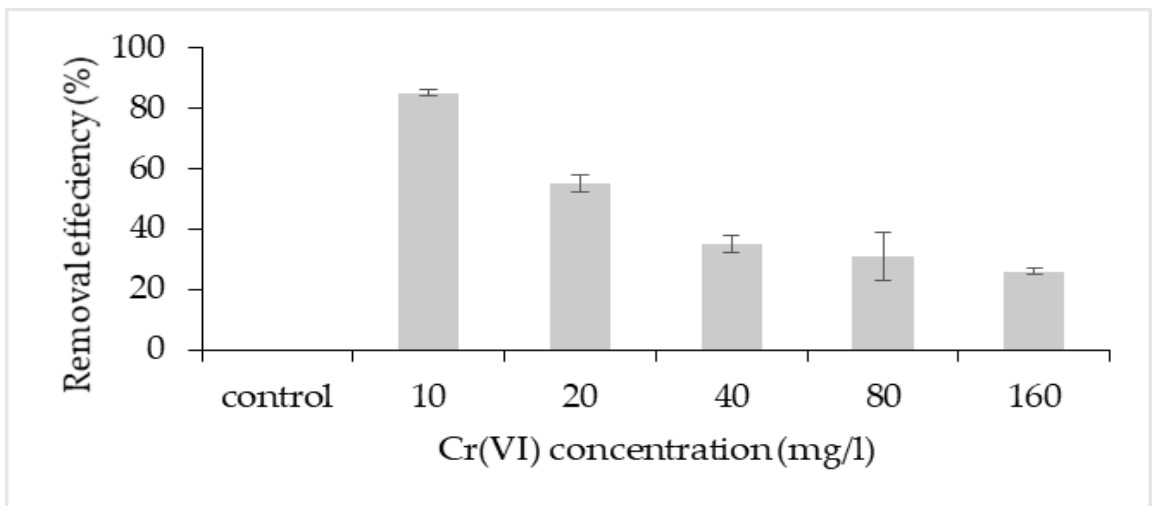

**Figure 3.** Cr(VI) removal efficiency at different initial Cr(VI) concentration.

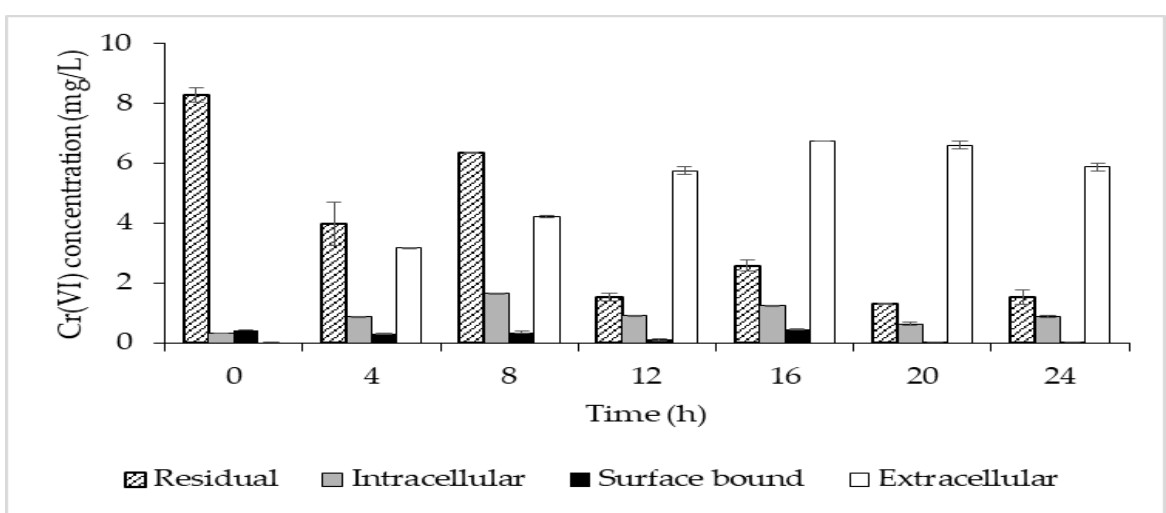

**Figure 4.** Time profile of Cr(VI) concentration in the broth (residual—patterned column), Cr(VI) accumulated in the cells (intracellular—grey column), Cr(VI) bound to the cell surface (surface-bound—black column) and Cr(VI) sequestered outside the cells (extracellular—clear column).

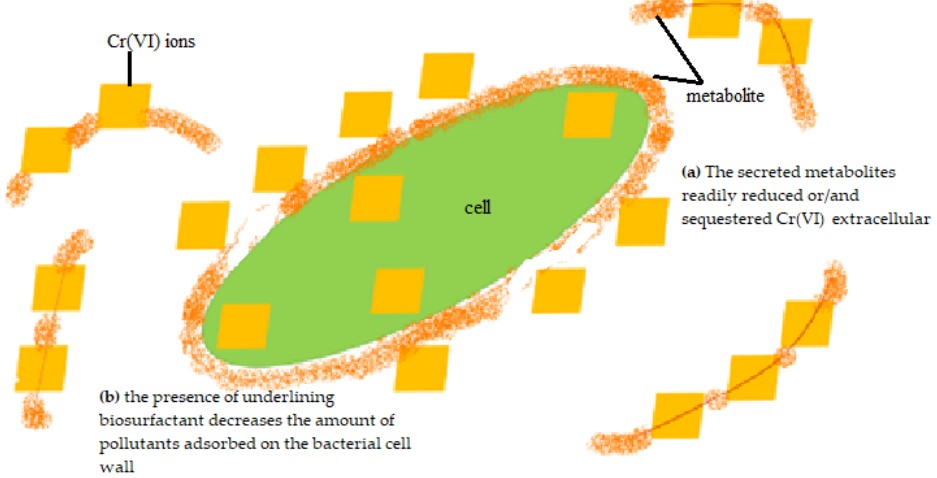

**Figure 5.** Proposed removal mechanisms of *P. aeruginosa* RW9.

FTIR analysis was conducted to determine the functional groups involved in the Cr(VI) binding (Figure 6). In the control sample, the first peak observed at 3296.20 cm$^{-1}$ was shifted to 3289.01 cm$^{-1}$ in cells with Cr(VI), which might be attributed to the stretching of the –NH bond of amino groups and the presence of hydroxyl (-OH) groups. The change in peak position in cells with Cr(VI) was due to the complexation of amino and hydroxyl groups with Cr(VI). The same observation was also confirmed by [31]. The presence of the asymmetric stretching of the C-H bond of the aliphatic chains –CH$_2$ and -CH$_3$ was confirmed at peak 2927.82 cm$^{-1}$ in the control, which was shifted to 2930.50 cm$^{-1}$ in Cr(VI)-treated cells. This shift in peak manifested the binding of Cr(VI) to the functional group, which acted as a nucleophile [19]. The weak stretching vibration that was detected at the wavelength of 1512 cm$^{-1}$ signified the presence of a carbonyl (C=O) group in both samples. However, Cr(VI) complexation was found to have more affinity to occur on the other side of the aliphatic chains, which resided by the amide groups. This was revealed by the shifting of amide I and amide II peaks at 1647.61 cm$^{-1}$ and 1537.31 cm$^{-1}$ to 1646.18 cm$^{-1}$ and 1540.46 cm$^{-1}$, respectively [32]. The role of the carboxylate (-COOH) functional group in Cr(VI) adsorption was also observed through the stretching of peak 1453.96 cm$^{-1}$ and 1233.65 cm$^{-1}$ to 1453.43 cm$^{-1}$ and 1230.95 cm$^{-1}$, respectively [33]. In addition, [34] reported that the bands in the region 1500–1200 cm$^{-1}$ also signified the deformation of the aliphatic chains –CH$_2$ and the angular deformation of C–C–H and H–C–O, which showed the complexation of Cr(VI) to the functional groups that have been discussed previously. The stretching of peak 1071.65 cm$^{-1}$ to 1072.44 cm$^{-1}$ confirmed the complexation of Cr(VI) to the phosphate functional group found in lipopolysaccharide, which was among the main cell membrane constituents of Gram-negative bacteria [35]. This FTIR finding confirmed the involvement of several functional groups that made up the cell membrane in the adsorption of Cr(VI), which was also reported by [36].

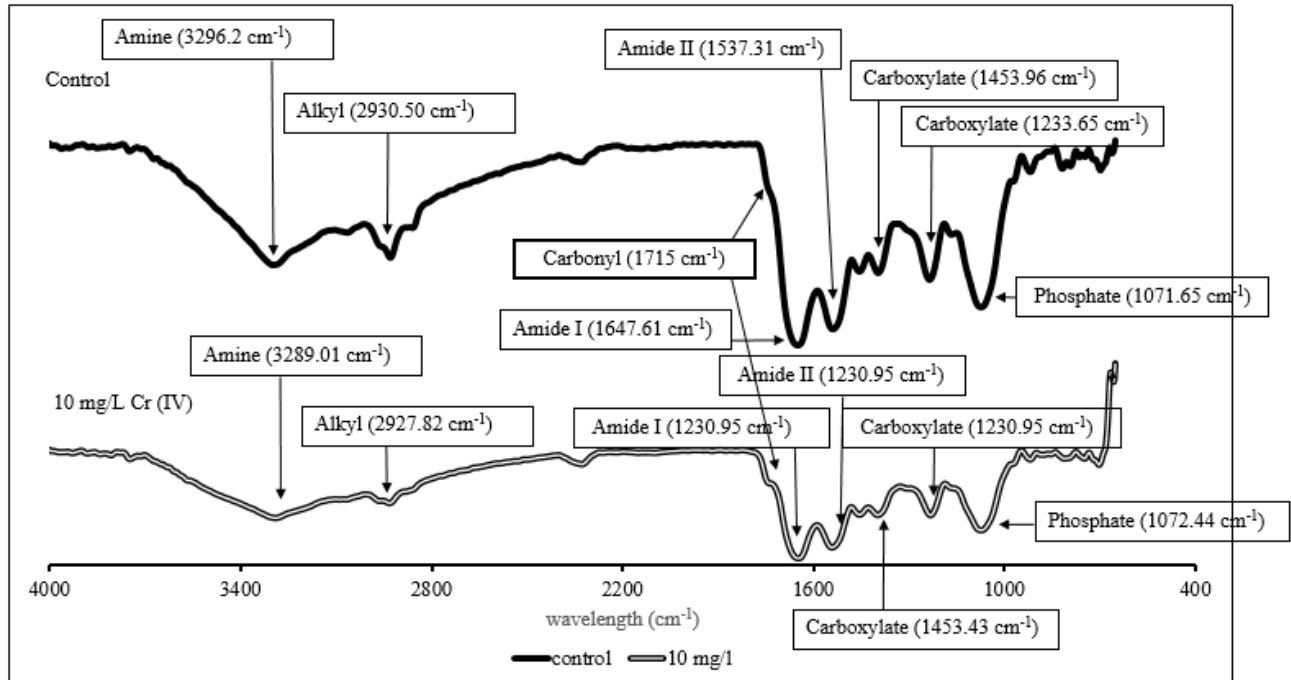

**Figure 6.** FTIR spectra of *P. aeruginosa* RW9 in NB without 10 mg/L of Cr(VI) as control (single-line), and NB with 10 mg/L of Cr(VI) (double-line) at 24 h of incubation.

### 3.3. Biosurfactant Quantification and Characterization

The orcinol test confirmed the presence of a biosurfactant as one of the metabolites that contributes to Cr(VI) extracellular sequestration (Figure 7). It was observed that the presence of Cr(VI) had slightly enhanced the production of biosurfactant relative to the con-

trol, which was also reported in previous studies by [37,38]. At the end of the fermentation process, 235 mg/L of biosurfactant was produced with 10 mg/L Cr(VI), slightly higher than the control (221 mg/L). In another report, 600 mg/L of biosurfactant was produced compared to control (200 mg/L) in Cd-treated cells [37], which is in accordance with the pattern of biosurfactant produced observed in this study. The increment was explained as being due to the expression of a rhlB gene that is enhanced in the presence of metals. rhlB is responsible for encoding the rhamnosyl transferase involved in the final step of L-rhamnosyl-3-hydroxydecanoyl-3-hydroxydecanoate synthesis [38]. In bioremediation, the biosurfactant was found to function in two ways: (1) as the cells' protective barrier, which prevents the penetration of the hazardous ion into the cells, and (2) helping in Cr(VI) removal through complexation formation, which reduced Cr(VI) to the less hazardous form of Cr(III). The FTIR-ATR was conducted to confirm the type of biosurfactant produced. The spectra obtained were found to exhibit a characteristic common to those of rhamnolipid, displaying the presence of carboxyl, carbonyl, ester and ether groups. These two functional groups (ester and carboxyl) signified the presence of fatty acids, which are the building blocks for the hydrophobic part of rhamnolipid [24]. It is also worth mentioning that there was slight peak shifting for each functional group detected, which might signify the role of biosurfactants in metal complexation.

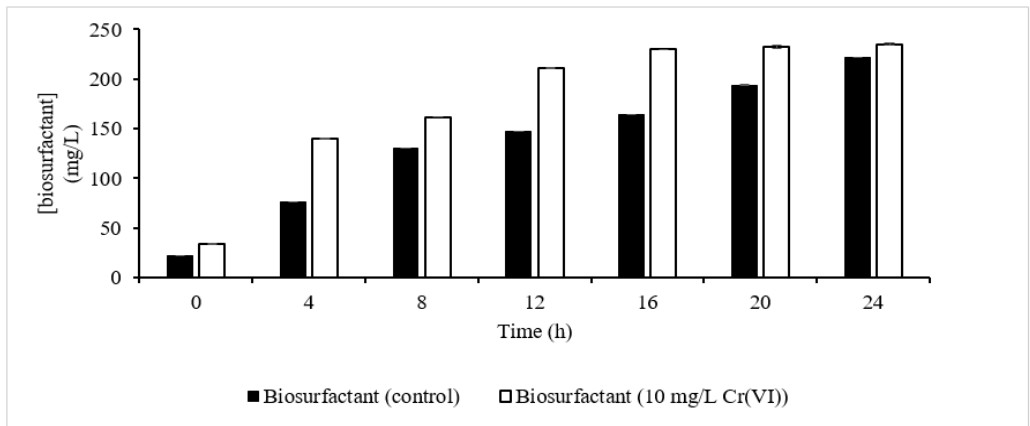

**Figure 7.** Biosurfactant synthesis profile. Biosurfactant synthesis in control experiment (dark column); biosurfactant synthesis in the presence of 10 mg/L Cr(VI) (clear column).

## 4. Conclusions

The present study showed that *P. aeruginosa* RW 9 was able to remove up to 85% of 10 mg/L Cr(VI). It was found that extracellular sequestration was the main Cr(VI) removal mechanism of this strain, which accounted for more than 50% of the total removal. The finding also suggested that Cr(VI) induced biosurfactant synthesis, and the biosurfactant was identified as a rhamnolipid. It showed that there was metal complexation formation considering the peak shifting recorded by the spectra. As future work, a detailed study will be conducted to elucidate the role of biosurfactants in Cr(VI) removal in *P. aeruginosa* RW 9.

**Supplementary Materials:** The following are available online at https://www.mdpi.com/article/10.3390/su13179797/s1, Figure S1: Steps to obtain samples in each treatment phase.

**Author Contributions:** Conceptualization, M.Z.M.Y.; methodology, F.M.A.; investigation, F.M.A. and A.F.A.; data curation, M.Z.M.Y.; writing— F.M.A.; writing—review and editing, H.A. and N.R.; visualization, T.M.; supervision, H.A.; N.R., T.M. and M.Z.M.Y.; resources, M.A.H.; funding acquisition, M.Z.M.Y.; All authors have read and agreed to the published version of the manuscript.

**Funding:** This research was funded by The Ministry of Education for the research fund FRGS/1/2019/TK10/UPM/02/1. Graduate Research Fund, School of Graduate Studies, Universiti Putra Malaysia.

**Institutional Review Board Statement:** Not applicable.

**Informed Consent Statement:** Not applicable.

**Acknowledgments:** The authors would like to thank Helmi Wasoh from UPM for providing the strain. Additionally, we acknowledge and thank the Ministry of Education for the research fund FRGS/1/2019/TK10/UPM/02/1 and Universiti Putra Malaysia for the Graduate Research Fund provided to the first author Fatini Mat Arisah.

**Conflicts of Interest:** The authors declare no conflict of interest.

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
