# Peer review of "Bacterial Resistance against Heavy Metals in Pseudomonas aeruginosa RW9 Involving Hexavalent Chromium Removal"

_sustainability, doi:10.3390/su13179797_

Round 1
Reviewer 1 Report
Title: Bacterial Resistance against Heavy Metal in Pseudomonas aeruginosa RW9 towards Hexavalent Chromium Removal
Abstract: This section is well written and captures the entire study.
Introduction:
I find this section also well written
Materials and Methods:
I find this section well-structured and scientifically valid. I will only suggest some description of the study area is provided.
Results and discussion: The results are well presented. However, I will suggest the discussion component is revised a bit to compare the study with similar ones.
The manuscript is well concluded and highlighting on possible future work was impressive.
In general, I find the study very important since conventional methods employed in the treatment of wastewater is proven to be a financial strain. Also, the use of microbial remediation will fit as an environmentally sustainable alternative hence will require more research attention.
I will also suggest some minor grammar corrections are done during proof reading. Aside this I find the paper very interesting, important and will surely appeal to readers.
Author Response
Reviewer 1
Manuscript ID: sustainability-1337426
Type of Manuscript: Article
Title: Bacterial Resistance against Heavy Metal in Pseudomonas aeruginosa Rw9
Thank you very much for reviewing the manuscript. we really appreciate your input and suggestion. The authors have revised the draft and made several amendments in response to the comments and suggestions. We appreciated the kind assistance given by the reviewers which help us to improve the quality and presentation of the manuscript.
Summary: We have revised our manuscript based on the reviewers’ comments. The comments from reviewers (bold) as well as responses were listed in the table below. The changes made in the manuscript were highlighted in yellow colour and justification for the amended sections were made in the same column. The indicated pages are based on the manuscript draft with marked changes. All changes are highlighted in yellow in the manuscript.
the detail response as attached below

Reviewer 2 Report
The submitted manuscript is well-prepared and interesting. Nevertheless, I have some comments and suggestions:
1) Line 79: It is not clear what is relationship between ability to degrade long-chain fatty acids and remediation of heavy metals. Please explain it in the text.
2) Line 100: The abbreviations like NB should be explained when they occured first time.
3) Line 161: The title says about extraction which is not mentioned in the paragraph below.
4) Lines 253-275: The IR signals identifiation is broad but the Authors focused on the confirmation of amides presence. However, I think that the authors should mentioned if the signals of groups like -OH, -C=O, which also are frequently present in biomolecules.
5) Section 3.3: It was assupted that the extracellular compounds are biosurfactants of rhamnolipid nature. In my opinion it is a little bit too far a priori statement. Not every P. aeruginosa strain produce rhamnolipids. Isn't it simple EPS composed of carbohydrates? Is this isolated compound possesed surface active properties?
Author Response
Thank you very much to all reviewers for reviewing the manuscript. The authors have revised the draft and made several amendments in response to the comments and suggestions. We appreciated the kind assistance given by the reviewers which help us to improve the quality and presentation of the manuscript.
We have revised our manuscript based on the reviewers’ comments. The comments from reviewers (bold) as well as responses were listed in the table below. The changes made in the manuscript were highlighted in yellow colour and justification for the amended sections were made in the same column. The indicated pages are based on the manuscript draft with marked changes. All changes are highlighted in yellow in the manuscript.

Round 2
Reviewer 2 Report
I thank the Authors for their kind responses. I think that now the manuscript is suitable for publiaction.